# Simple Synthesis of Molybdenum Carbides from Molybdenum Blue Nanoparticles

**DOI:** 10.3390/nano11040873

**Published:** 2021-03-30

**Authors:** Natalia Gavrilova, Maria Myachina, Victor Nazarov, Valery Skudin

**Affiliations:** 1Department of Colloid Chemistry, Faculty of Natural Sciences, D. Mendeleev University of Chemical Technology of Russia, Miusskaya sq., 9, 125047 Moscow, Russia; mmyachina@muctr.ru (M.M.); nazarov@muctr.ru (V.N.); 2Department of Chemical Technology of Carbon Materials, Faculty of Petroleum Chemistry and Polymers, D. Mendeleev University of Chemical Technology of Russia, Miusskaya sq., 9, 125047 Moscow, Russia; skudin@muctr.ru

**Keywords:** molybdenum carbide, molybdenum blue, nanoclusters, dispersion, sol-gel method

## Abstract

In recent years, much attention has been paid to the development of a new flexible and variable method for molybdenum carbide (Mo_2_C) synthesis. This work reports the applicability of nano-size clusters of molybdenum blue to molybdenum carbide production by thermal treatment of molybdenum blue xerogels in an inert atmosphere. The method developed made it possible to vary the type (glucose, hydroquinone) and content of the organic reducing agent (molar ratio R/Mo). The effect of these parameters on the phase composition and specific surface area of molybdenum carbides and their catalytic activity was investigated. TEM, UV–VIS spectroscopy, DTA, SEM, XRD, and nitrogen adsorption were performed to characterize nanoparticles and molybdenum carbide. The results showed that, depending on the synthesis conditions, variants of molybdenum carbide can be formed: α-Mo_2_C, η-MoC, or γ-MoC. The synthesized samples had a high specific surface area (7.1–203.0 m^2^/g) and meso- and microporosity. The samples also showed high catalytic activity during the dry reforming of methane. The proposed synthesis method is simple and variable and can be successfully used to obtain both Mo_2_C-based powder and supports catalysts.

## 1. Introduction

In recent years, the development of catalysts based on transition metal carbides has become increasingly important in chemical technology. Transition metal carbides show high hardness, thermal stability, and catalytic properties similar to Pt group metals [1,2]. One such promising catalyst is molybdenum carbide, Mo_2_C. Molybdenum show high selectivity and activity in many reactions, such as isomerization, ammonia synthesis, water–gas shift reaction, hydrocarbon reforming, and H_2_ production [3,4,5,6,7,8,9]. Molybdenum carbides also demonstrate resistance to poison [10,11,12], so they are considered an alternative to platinum-group catalysts.

The most widely used method of synthesizing molybdenum carbides is the temperature-programmed reaction (TPRe), which involves gas–solid reactions. Molybdenum oxides are used as a molybdenum precursor and a mixture of hydrocarbon gases (C_n_H_2n+2_/H_2_) as a carbon source [13,14,15,16,17,18,19,20]. Studies on molybdenum carbide synthesis via the TPRe have specified various conditions: the rate of temperature change, the final temperature of the process, the carbon source, and the H_2_/C_n_H_2n+2_ ratio. The choice of activating mixtures affects the specific surface area, morphology, and particle size of the resulting carbide, as well as its catalytic activity [19,20]. This method usually forms β-Mo_2_C. Preparing other phases, such as monocarbides (MoC), however, requires a complex synthesis procedure involving preliminary nitridation of molybdenum oxides, followed by the carbidation of molybdenum nitride, pretreatment with hydrogen, or treatment of molybdenum oxides at high pressure [21,22].

Despite its advantages, the TPRe has a few limitations. First, it involves the use of explosive mixtures (hydrocarbon and hydrogen), which requires certain safety measures. Second, hydrocarbon interaction inevitably forms carbon, which is deposited as a polymer film on the catalyst surface, blocking active sites [23]. In addition, the process of obtaining supported catalysts is complex. At the first stage, it is necessary to use molybdenum salt solutions for the impregnation of supports, followed by calcination, and then the MoO_3_ supported catalysts are carburized using the gas mixtures described above [24,25].

Among the various methods for molybdenum carbide synthesis, soft chemistry, which provides an opportunity to obtain molybdenum carbides at relatively low temperatures or under mild conditions, is especially important. In most cases, liquid-phase systems containing molybdenum salts are a molybdenum source and organic compounds are a carbon source. The resulting precipitate or gel is subsequently sintered in an inert atmosphere. For example, in [26,27,28], molybdenum carbide was produced using ammonium heptamolybdate or pentachloride and glucose, sucrose, alanine, or urea. In [26], the authors emphasized that a novel method (sol-gel-like synthesis) of transition metal carbide production exists and is in high demand. The observed sol-gel transition opens up the possibility of obtaining various Mo_2_C-based materials such as fibers, coatings, and supported catalysts.

Molybdenum polyoxometalate structures, represented by molybdenum oxide clusters of various shapes and sizes, are of great interest in this area of research. Molybdenum blue nanoparticles are dispersions of molybdenum oxide clusters in which molybdenum is present in +5 and +6 oxidation states. These compounds can be obtained by the reduction of molybdates at pH ≤ 3 [29,30,31]. Organic compounds can be used as reducing agents, which will act as a carbon source in molybdenum carbide synthesis.

For many years, molybdenum blue nanoparticles were considered interesting exclusively in inorganic chemistry. However, they can also be used in the preparation of different catalysts due to the peculiarities of their synthesis and structure. Their small size, monodispersity, and the presence of partially reduced molybdenum make them promising precursors for molybdenum carbide synthesis.

Oxide clusters in the molybdenum blues are so large that they can be considered as hydrosols. This makes them convenient for the sol-gel technology using all of the advantages of the sol-gel method: the possibility of deposition on porous and nonporous supports, morphology control, and the porosity and thickness of the catalytic layer. 

A molybdenum blue dispersion synthesized with ascorbic acid was successfully used to prepare Mo_2_C [32]. The molar ratio (R/Mo) is the main parameter that affects the morphology, pore structure, and phase composition of molybdenum carbide. At a low R/Mo molar ratio, monophase mesoporous β-Mo_2_C is formed, while increasing the reducing agent content leads to the formation of a second phase, η-MoC. 

Stable molybdenum blue dispersions from the viewpoint of aggregative and chemical stability can also be obtained by using other reducing agents such as glucose [33] and hydroquinone [34], and it would be interesting to study the possibility of using these dispersions for molybdenum carbide synthesis. 

In this study we synthesized molybdenum carbides using molybdenum blue and organic reducing agents (glucose and hydroquinone) and investigated the effect of the type and content of the reducing agent on the morphology, phase composition, and surface area of the synthesized molybdenum carbides. The results open up the possibility of a simple route of sol-gel synthesis for the molybdenum carbides and catalysts based on them.

## 2. Materials and Methods

### 2.1. Materials

Ammonium heptamolybdate (NH_4_)_6_Mo_7_O_24_∙4H_2_O, glucose (C_6_H_12_O_6_), hydroquinone (C_6_H_6_O_2_), and hydrochloric acid (HCl) were used as precursors for molybdenum blue dispersion synthesis. All chemicals used were laboratory grades (CT Lantan, Moscow, Russia).

### 2.2. Molybdenum Blue Synthesis

A strictly defined amount of an organic reducing agent was added to ammonium heptamolybdate solution (0.07 M) with vigorous stirring, and then hydrochloric acid was added until the pH reached 2.0. The reaction was carried out at room temperature.

The synthesis conditions were as follows: molar ratio R/Mo = 4.0–7.0, molar ratio of hydrochloric acid and molybdenum H/Mo = 0.5 for glucose and R/Mo = 4.0–9.0, and molar ratio H/Mo = 3.0 for hydroquinone. 

To synthesize 50 mL of molybdenum blue dispersion using glucose, 10.6 mL of ammonium heptamolybdate solution (0.047 M), 0.3 mL of hydrochloric acid (2.46 M), and 2.52–4.41 g of glucose powder (R/Mo = 4.0–7.0) were used. To synthesize 50 mL of molybdenum blue dispersion using hydroquinone, 10.6 mL of ammonium heptamolybdate solution (0.047 M), 4.2 mL of hydrochloric acid (2.46 M), and 1.54–3.47 g of hydroquinone powder (R/Mo = 4.0–9.0) were used. 

These conditions were set based on previous experiments to form stable (aggregative and chemical) molybdenum blue dispersions [33,34].

### 2.3. Molybdenum Blue Dispersions Characterization

TEM was performed using a LEO 912 AB Omega microscope (Carl Zeiss, Jena, Germany) to determine the particle size distribution. XPS spectra were recorded on an ESCA+X-ray photoelectron spectrometer (OMICRON Nanotechnology, Taunusstein, Germany). UV–VIS spectra were recorded using a Leki SS2110 UV scanning spectrophotometer (MEDIORA OY, Helsinki, Finland). 

### 2.4. Molybdenum Carbide Synthesis

Xerogels were obtained by drying molybdenum blue dispersions at room temperature. Molybdenum carbides were prepared by thermal decomposition of the xerogels; thermal treatment was carried out in a N_2_ atmosphere at 900 °C.

For further experiments, the following samples were synthesized: G-4, G-5, G-6, and G-7 (reducing agent, glucose; number, molar ratio R/Mo) and H-4, H-5, H-6, H-7, H-8, and H-9 (reducing agent, hydroquinone; number, molar ratio R/Mo).

### 2.5. Molybdenum Carbide Characterization

Thermal analysis (DTA) of xerogels was performed using an SDT Q600 thermal analyzer (TA Instruments, New Castle, DE, USA) in inert atmosphere. XRD patterns were recorded using a Rigaku D/MAX 2500 diffractometer (Rigaku Corporation, Tokyo, Japan) with CuKα radiation at a scanning step of 0.02 and a scanning range (2θ) of 20°–90°. The specific surface area (S_a_) of the samples was determined using Gemini VII (Micromeritics, Norcross, GA, USA) by applying the standard Brunauer–Emmett–Teller (BET) method. The total specific pore volume (ΣV) was determined at a maximum relative pressure of 0.995, and the pore size distribution was calculated using BJH (mesopores) and Horvath–Kawazoe (micropores) methods. The volume of micropores was determined using the Dubinin–Radushkevich equation and a t-plot.

### 2.6. Catalytic Tests 

The catalytic activity of the synthesized molybdenum carbide samples (G-7, H-4) in dry reforming of methane was evaluated using a fixed-bed reactor at atmospheric pressure and temperature 800 °C–870 °C. The feed gas mixture comprised an equimolar mixture of CH_4_ and CO_2_. Gas compositions were monitored using a Crystall 5000 gas chromatograph (CJSC SKB Chromatec, Yoshkar-Ola, Russia) equipped with two thermal conductivity detectors and chromatographic columns (HayeSep R 80/100, NaX 60/80).

The parameters of catalytic activity (the rate constant k^900^ and specific rate constant k_s_^900^) were calculated according to the work [32].

## 3. Results

### 3.1. Molybdenum Blue Dispersion Characterization

Molybdenum blue dispersions were synthesized by the reduction of ammonium heptamolybdate in an acidic medium. Glucose and hydroquinone were used as reducing agents due to the possibility of using them as a carbon source in subsequent further molybdenum carbide synthesis. 

Figure 1 shows TEM micrographs of synthesized molybdenum blue dispersions. The dispersed phase was represented by particles less than 5 nm in size. The most probable particle diameter calculated from particle size measurements was around 3 nm. 

Figure 2 shows the absorption spectra of the synthesized molybdenum blue dispersions. The two characteristic absorption bands in the visible part of the spectrum (750 nm) and the near-IR region (1050 nm) indicated not only the presence of reduced molybdenum but also the toroidal shape of particles of the dispersed phase (molybdenum oxide clusters) [29,35,36].

### 3.2. Synthesis of Molybdenum Carbides from Molybdenum Blue Dispersions

To determine the temperature required to obtain molybdenum carbides, thermogravimetric analysis of molybdenum blue xerogels synthesized using glucose and hydroquinone was performed (Figure 3).

The sample weight loss occurred stepwise in several stages. The first stage occurred at 100 °C and 180 °C and was associated with the removal of free and bound water. Simultaneously, several endothermic effects were recorded on the DTA curves. At temperatures above 350 °C, the decomposition of organic substances and ammonium chloride began. In this region, a diffuse exothermic effect was observed, with a maximum at ~350 °C.

The last stage of mass change began at 680 °C–750 °C. These changes indicated the forming of carbide from carbon and molybdenum oxides, accompanied by a slight endothermic effect.

To analyze the changes in the solid product of xerogel decomposition, a series of samples were prepared, calcined at different temperatures, and subjected to X-ray phase analysis (Figure 4). 

The diffractograms showed two diffuse reflections for molybdenum blue xerogels synthesized using glucose (Figure 4a), calcined at temperatures of 600 °C and below, corresponding to amorphous carbon. Reflections of another phase (molybdenum compounds) began to appear at temperatures above 600 °C. These reflections (26.0, 37.0, 53.1, 60.2) corresponded to monoclinic molybdenum dioxide, MoO_2_ (32-0671). The carbon and molybdenum dioxide phases were present in the diffraction patterns up to temperatures of 800 °C–830 °C. At 900 °C, characteristic reflections of hexagonal α-Mo_2_C (35-0787) appeared.

In contrast to the series of samples synthesized using glucose, there was practically no amorphous carbon in the diffractograms of the series of samples synthesized using hydroquinone, since the excess reducing agent present in the xerogels sublimed (Figure 4b). Therefore, the reflections of molybdenum compounds were clearly visible. At temperatures of 500 °C–700 °C, there was one phase of molybdenum dioxide, MoO_2_ (32-0671). An increase in temperature to 800 °C led to the formation of molybdenum carbide α-Mo_2_C (35-0787). For samples obtained by calcination at 900 °C, the presence of two phases of α-Mo_2_C and γ-MoC (45-1015) was observed. Analysis of the literature data showed that the solid–liquid phase method has not yet succeeded in obtaining γ-MoC or its mixture with α-Mo_2_C [9].

To determine the effect of the reducing agent content on the phase composition of molybdenum carbide obtained from molybdenum blue dispersions, a series of samples with the molar ratio R/Mo = 4–9 was prepared. The calcination temperature of the samples was 900 °C. The results are shown in Figure 5.

Samples with a low reducing agent content (glucose, R/Mo = 4–5) were characterized by the presence of one phase of molybdenum carbide, β-Mo_2_C (35-0787). The sample with R/Mo = 5 was characterized by the maximum intensity of reflections of the β-Mo_2_C phase and the lowest intensity of reflections of free carbon (see Figure 5a). This indicated the completeness of the carbide formation from its precursors. An increase in the reducing agent content led to the appearance of η-MoC and an increase in free amorphous carbon. The results are in good agreement with the phase diagram, according to which η-MoC is formed in the presence of the excess carbon [37].

Diffuse reflections in the diffractogram indicate high dispersion of the material, which, in this case, could be provided for uniform distribution of molybdenum carbide particles over a carbon matrix.

The phase composition of the samples synthesized using hydroquinone was represented by two compounds: α-Mo_2_C (35-0787) and γ-MoC (45-1015) (Figure 5b). At the lowest reducing agent content (R/Mo = 4), the predominant phase was α-Mo_2_C, and an increase in the molar ratio R/Mo led to an increase in the γ-MoC content. To establish the structure of free carbon presented in samples, Raman spectra were used (Figure 6).

Figure 6 shows that the spectrum in the region of 500–2500 cm^−1^ contains two peaks, 1591 and 1336 cm^−1^, characteristic of the disordered structure of carbon materials. A narrow and clear peak at 1582 cm^−1^ is characteristic of a single crystal of graphite (G-line). For disordered structures, this line is blurred and one more line appears at lower values of the shift (D-line) [38]. The intensity ratio of these lines, I_D_/I_G_, enables calculating the degree of ordering in the carbon material. In the studied samples, this ratio was 1.15 and 1.20 for glucose and hydroquinone, respectively, indicating a low ordering in the studied carbon material. Thus, molybdenum carbide contains amorphous carbon, as confirmed by XRD data and Raman spectroscopy.

### 3.3. Molybdenum Carbide Characterization

The morphology of the synthesized molybdenum carbide samples was analyzed using scanning electron microscopy. Figure 7 shows micrographs of Mo_2_C particles obtained by heat treatment of molybdenum blue xerogels in an inert atmosphere at 900 °C. 

Figure 7a,b show that the molybdenum carbide samples (G-5) contained sharp particles with clear edges. On the fragment surface, smaller, irregular-shaped particles were observed. Highly dispersed particles of molybdenum carbide were distributed in the carbon matrix, which led to broad reflections of molybdenum carbide in the diffraction patterns of the samples synthesized using glucose. The predominant size of primary molybdenum carbide particles enclosed in the carbon matrix was 400 nm. It should be noted that the same morphology was observed for all investigated samples of carbides, synthesized using glucose (G-4–G-9).

Figure 7c,d show photomicrographs of molybdenum carbide obtained by heat treatment of a molybdenum blue xerogel synthesized using hydroquinone. Most of the objects in the field of view were aggregates of primary molybdenum carbide particles with irregular shape and diameter not more than 1 μm. In contrast to samples obtained with excess glucose, the sample under consideration contained a small number of fragment-like particles of amorphous carbon. An increase in the carbon content in samples H-5/H-9 leads to an insignificant change in the morphology of the samples according to SEM results.

To determine the characteristics of the porous structure, we performed low-temperature adsorption of N_2_; the adsorption and desorption isotherms are shown in Figure 8.

The adsorption isotherms of molybdenum carbide samples (G-5, H-4) belonged to type IV, according to the Brunauer classification, which is characteristic of polymolecular adsorption—capillary condensation in mesopores. The shape of the hysteresis loop was of type H3, corresponding to the presence of slit pores. In the initial section of the isotherm, we observed a sharp increase in adsorption, indicating the presence of micropores. The adsorption isotherm presented in the coordinates of the t-method had three characteristic sections corresponding to the adsorption in micro-, meso-, and macropores.

The pore size distribution calculated by the BJH method (for mesopores) and the Horvath–Kawazoe method (for micropores) is shown in Figure 9. The sample had a rather narrow pore size distribution, and the prevailing diameters (slit pore width) were 4 and 1 nm.

Molybdenum carbide samples obtained by heat treatment of molybdenum blue xerogels synthesized using glucose at different R/Mo molar ratios were investigated in a similar way. For all investigated samples, isotherms similar to those shown in Figure 8 were obtained.

The characteristics of the porous structures of the samples are shown in Table 1.

The specific surface area (total and internal) and pore volume of micropores increased with an increase in the reducing agent content. The main contributor to these characteristics was amorphous carbon.

Thus, the synthesized molybdenum carbide samples have a bimodal porous structure. Microporosity is due to the presence of amorphous carbon, and mesoporosity with a predominant diameter around 4 nm corresponds to the structure of molybdenum carbide, in which pores are formed due to sintering of primary particles. A similar porous structure is observed in the synthesis of molybdenum carbide using ascorbic acid as a carbon source [32].

### 3.4. Catalytic Activity of Molybdenum Carbides

Molybdenum carbide samples obtained by thermal decomposition of molybdenum blue xerogels, characterized by different phase compositions (Table 2) were tested as a catalyst during the dry reforming of methane. The reaction was carried out in a temperature range of 800 °C–870 °C. H_2_, CO, and H_2_O were found among the reaction products. 

The temperature dependencies of the conversions of starting materials (CH_4_ and CO_2_) are shown in Figure 10, in addition to the H_2_/CO ratio, which indicates the selectivity of the catalysts under study. The catalyst obtained from molybdenum blue synthesized using hydroquinone had a higher catalytic activity at 870 °C, a conversion close to 90% was observed, with the highest H_2_/CO molar ratio.

To compare the obtained results, based on kinetic experiments, we determined the rate constant and specific rate constant per unit surface of the catalyst (Table 2). The molybdenum carbide sample, represented by two phases (α-Mo_2_C and γ-MoC), was more active. The sample with a different phase composition (α-Mo_2_C and η-MoC) has the specific rate constant two-times lower. 

## 4. Discussion

The traditional method of producing molybdenum carbides is the TPRe based on the effect of a gas mixture (C_n_H_2n+2_/H_2_) on MoO_3_. Using CH_4_/H_2_ and C_2_H_6_/H_2_ mixtures leads to the formation of one phase of β-Mo_2_C [17,37,38,39,40]. The formation of monocarbides (MoC) is possible using hydrocarbons with a long chain of aromatic compounds [9,17]. There is practically no literature on α-Mo_2_C synthesis by the TPRe.

This study presented the results of molybdenum carbide synthesis by a simple method based on obtaining molybdenum blue dispersions and subsequently drying and heat-treating them in an inert atmosphere. By changing the reducing agent during molybdenum blue dispersion synthesis, it is possible to form different phases of molybdenum carbides. This synthesis method is flexible and scalable and can be considered an alternative to the traditional TPRe.

In this paper, we reported molybdenum carbide synthesis using glucose and hydroquinone. The phase composition of molybdenum carbides depends on the type and content of the organic reducing agent. When glucose is used, α-Mo_2_C and η-MoC are formed, and the ratio between them is determined by the amount of glucose: with minimum glucose, monophase samples are formed (Mo_2_C), and increasing the glucose content increases the amount of η-MoC. When hydroquinone is used, the presence of two phases is observed in the entire investigated range of the R/Mo molar ratio: α-Mo_2_C and γ-MoC. It should be noted that in our previous study it was shown that the use of ascorbic acid leads to the formation of a mixture of Mo_2_C and η-MoC carbides, as well as when glucose is used [32].

The chosen strategy for the synthesis of carbides refers to the method of obtaining a solid–liquid phase, in which organic compounds are used as a carbon source [9]. Table 3 summarizes the main properties of molybdenum carbide synthesized using a different liquid phase carbon source. This approach allows varying the content of the organic reducing agent, as in the case of the urea “solution” route, which leads to a change in the phase composition of the resulting carbide and a change in the specific surface area from 12 to 22 m^2^/g [26]. However, in the case of using urea, molybdenum nitrite is also formed in addition to molybdenum carbide. 

Comparing our results with previous studies with similar synthesis method, using a molybdenum salt and a liquid-phase carbon source (5 m^2^/g sucrose) [27], we found that, when using molybdenum blue dispersions as precursors, molybdenum carbides with a higher surface area (7.1–203 m^2^/g) are formed.

For the first time, samples containing γ-MoC were obtained and their activity during the carbon dioxide conversion of methane was established. The molybdenum carbide samples obtained in this study show high catalytic activity in the DRM reaction. Samples representing a mixture of phases have greater activity and selectivity in this reaction.

## 5. Conclusions

In this work, we developed a simple method for obtaining molybdenum carbides using dispersions of molybdenum blue and organic compounds (glucose and hydroquinone) as a carbon source. The ability to vary the type and content of the organic reducing agent makes it possible to synthesize molybdenum carbide of various phase composition and porous structure.

It has been shown that when glucose is used, α-Mo_2_C and η-MoC are formed, while when using hydroquinone, α-Mo_2_C and γ-MoC are formed.

The porous structure of the samples is represented by meso- and micropores, the specific surface area is in the range of 7.1–203 m^2^/g.

Samples of carbides were tested in the reaction of dry reforming of methane in the temperature range 800–890 °C. It was shown that a sample of a mixture of molybdenum carbides α-Mo_2_C and γ-MoC exhibits the highest catalytic activity.

The developed method for the synthesis of molybdenum carbides makes it possible to obtain bulk and supported catalysts based on molybdenum carbide by the sol-gel method.

## Figures and Tables

**Figure 1 nanomaterials-11-00873-f001:**
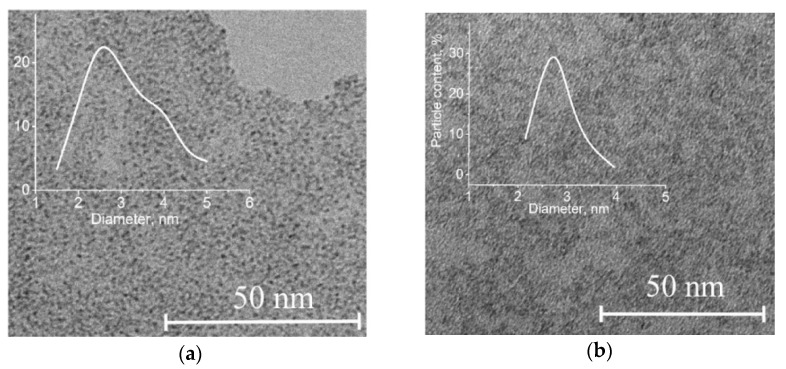
TEM images of particles and particle size distribution of molybdenum blue dispersions synthesized using various reducing agents: (**a**) glucose and (**b**) hydroquinone.

**Figure 2 nanomaterials-11-00873-f002:**
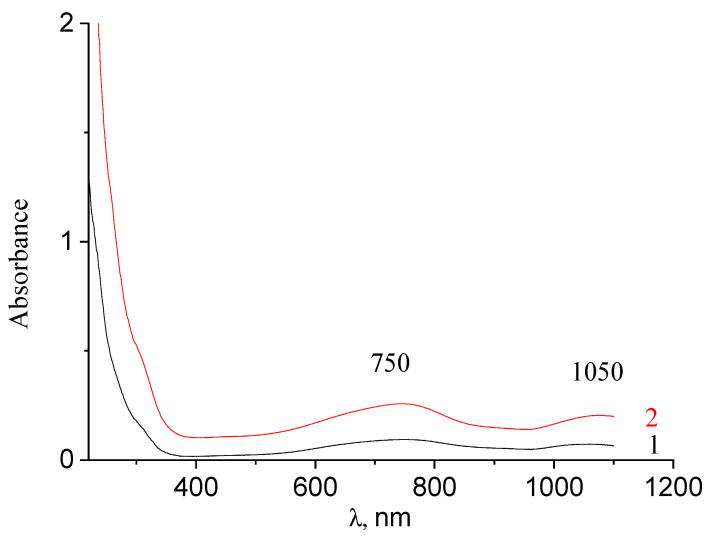
Electronic absorption spectra of molybdenum blue nanoparticles synthesized using various reducing agents: glucose (1) and hydroquinone (2).

**Figure 3 nanomaterials-11-00873-f003:**
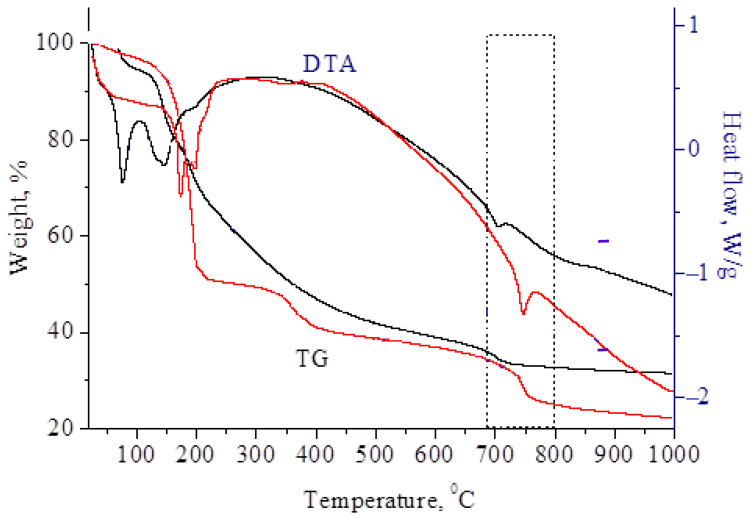
TG and DTA curves of molybdenum blue xerogels synthesized using glucose (black lines) and hydroquinone (red lines).

**Figure 4 nanomaterials-11-00873-f004:**
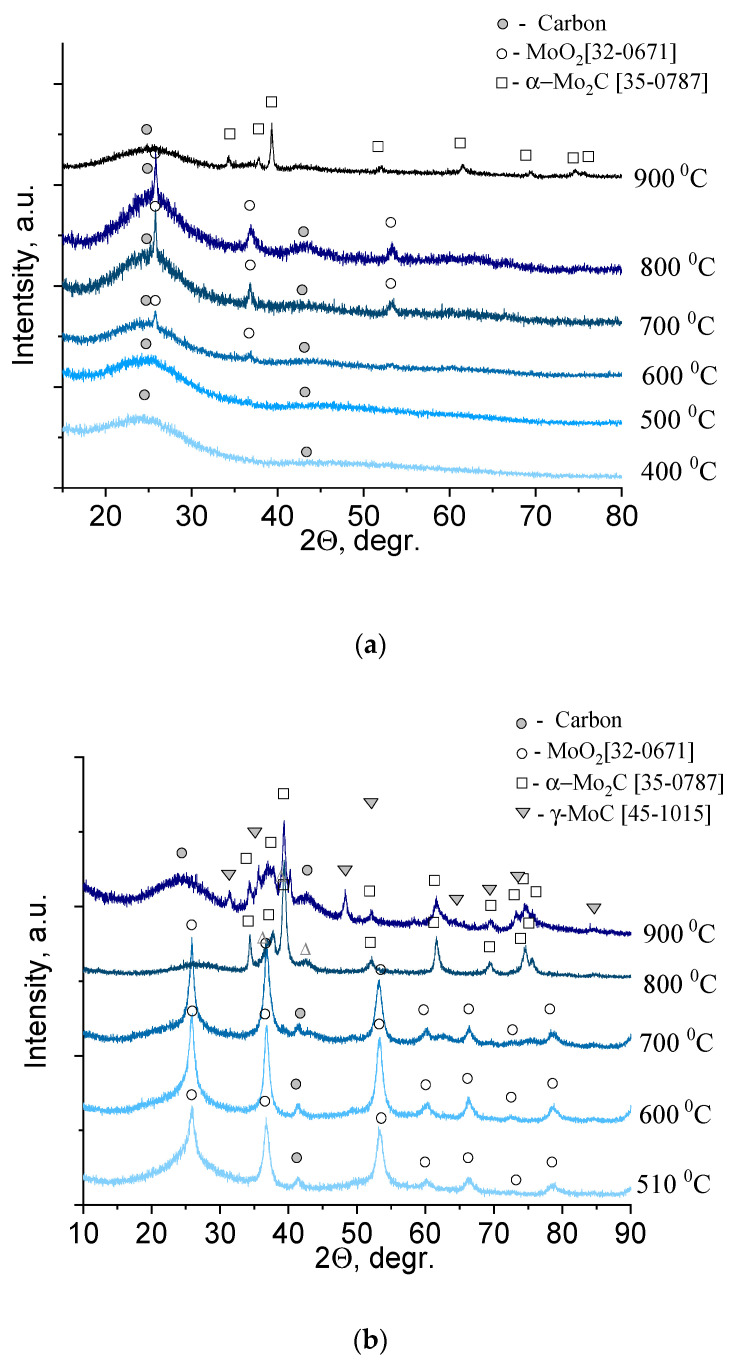
XRD pattern of molybdenum blue xerogels synthesized using (**a**) glucose (G-4, R/Mo = 4.0) and (**b**) hydroquinone (H-4, R/Mo = 4.0), calcined at different temperatures in an inert N_2_ atmosphere.

**Figure 5 nanomaterials-11-00873-f005:**
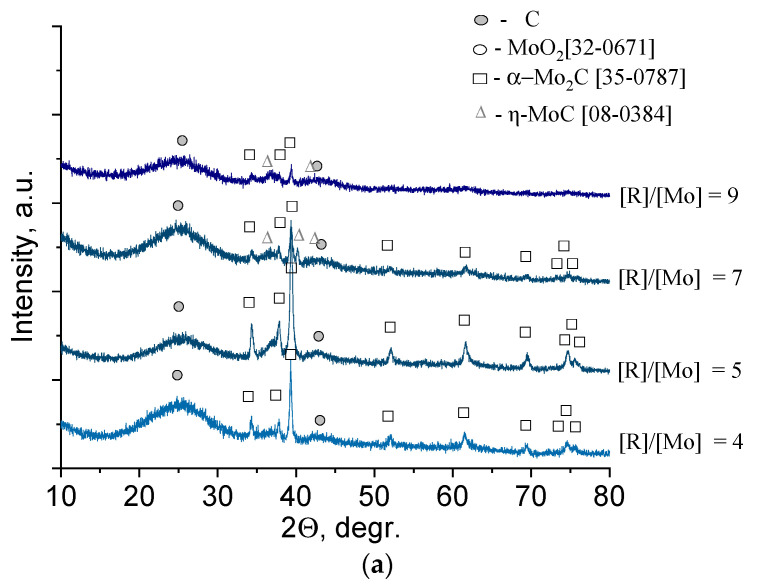
XRD pattern of molybdenum blue xerogels synthesized using (**a**) glucose and (**b**) hydroquinone with different R/Mo molar ratios calcined at 900 °C in an inert N_2_ atmosphere.

**Figure 6 nanomaterials-11-00873-f006:**
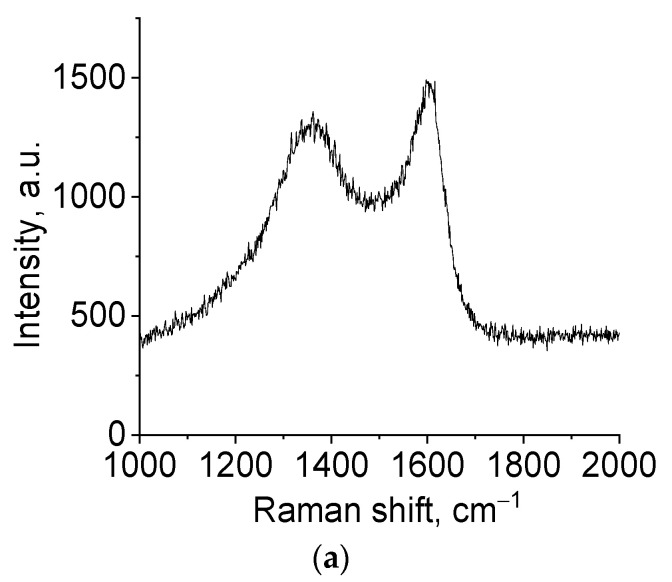
Raman spectra of a sample of molybdenum blue xerogels obtained using (**a**) glucose (G-7) and (**b**) hydroquinone (H-4) calcined at 900 °C in inert N_2_ atmosphere.

**Figure 7 nanomaterials-11-00873-f007:**
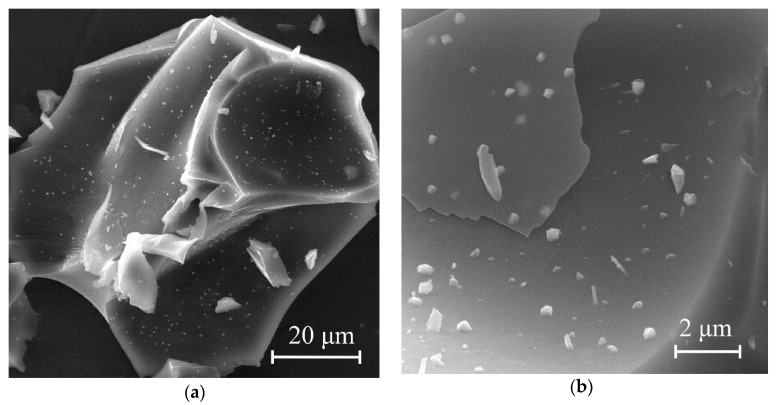
Micrographs of Mo_2_C obtained by heat treatment of a molybdenum blue xerogel synthesized using (**a**,**b**) glucose (G-5, R/Mo = 5) and (**c**,**d**) hydroquinone (H-4, R/Mo = 4).

**Figure 8 nanomaterials-11-00873-f008:**
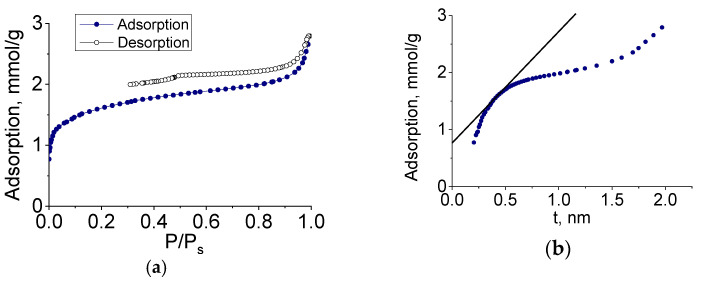
(**a**,**b**) Nitrogen adsorption isotherms and (**c**,**d**) t-plots of Mo_2_C synthesized from molybdenum blue xerogels (calcined at 900 °C in an inert atmosphere) using glucose (G-5, R/Mo = 5) (**a**,**b**) and hydroquinone (H-4, R/Mo = 4) (**c**,**d**).

**Figure 9 nanomaterials-11-00873-f009:**
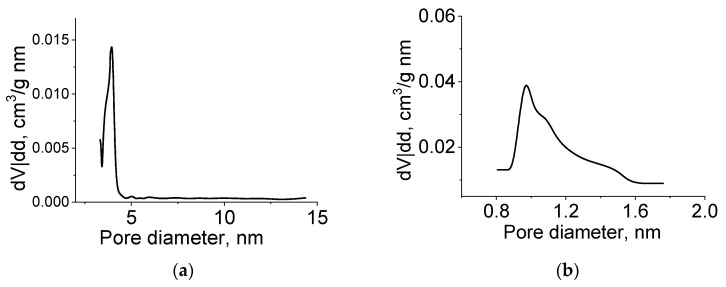
Meso- (**a**,**c**) and micropore (**b**,**d**) size distribution of molybdenum carbide synthesized from molybdenum blue xerogels using (**a**,**b**) glucose (G-5, R/Mo = 5) and (**c**,**d**) hydroquinone (H-4, R/Mo = 4), calcined at 900 °C in an inert atmosphere.

**Figure 10 nanomaterials-11-00873-f010:**
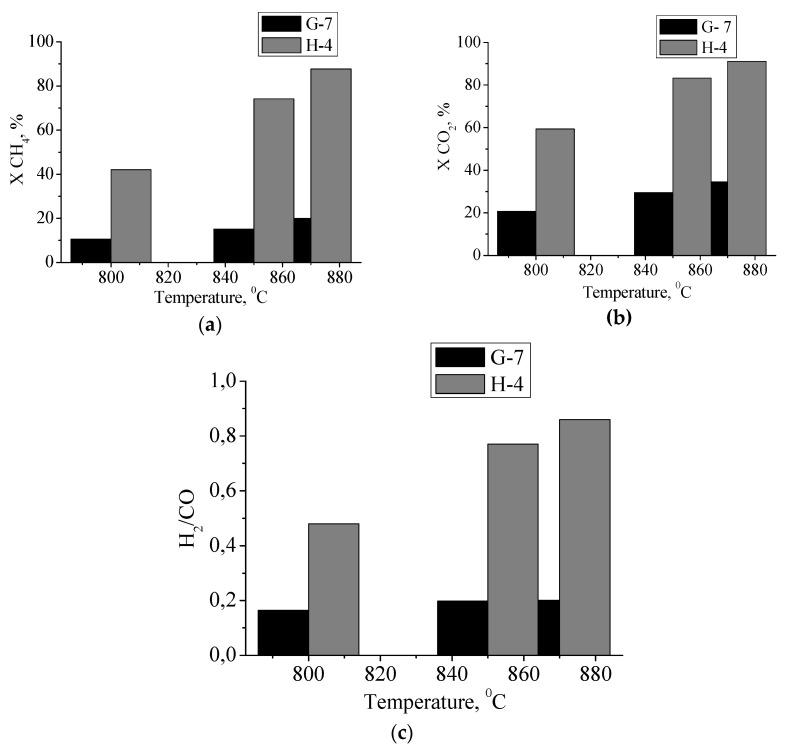
Effect of the phase composition of the catalyst on (**a**) CH_4_ conversion, (**b**) CO_2_ conversion, and (**c**) H_2_/CO ratio for dry reforming of methane. For all experiments, the flow rate of the gas mixture was 30 cm^3^/min and the contact time was 0.6 s.

**Table 1 nanomaterials-11-00873-t001:** Characteristics of the porous structure of Mo_2_C synthesized from molybdenum blue xerogels using glucose and hydroquinone with different R/Mo molar ratios, calcined at 900 °C in an inert atmosphere.

Sample	Molar Ratio (R/Mo)	Parameters					
		S_a_ (BET), m^2^/g	∑V, cm^3^/g	V_meso_ (BJH), cm^3^/g	V_0_ (DR), cm^3^/g	V_t_, (t-plot) cm^3^/g	d_p_, nm
Glucose							
G-4	4.0	7.1	0.011	0.007	0.003	0.001	4; 1
G-5	5.0	130.9	0.097	0.036	0.052	0.026	4; 1
G-7	7.0	174.9	0.079	0.001	0.071	0.046	4; 1
Hydroquinone							
H-4	4.0	203.0	0.150	0.079	0.082	0.043	4; 1

S_a_—specific surface area calculated using BET equation, ΣV—the total specific pore volume, V_meso_—mesopore volume, V_0_—the micropore volume determined based on the Dubinin–Radushkevich equation. V_t_—value of micropore volume calculated using the t-plot de Boer method, d_p_—predominant pore size.

**Table 2 nanomaterials-11-00873-t002:** Catalytic activity of Mo_2_C synthesized from molybdenum blue xerogels using glucose (R/Mo = 7) and hydroquinone (R/Mo = 4).

Sample	Reducing Agent, (R/Mo)	Parameters			
		Phase Composition	S_a_ (BET) *, m^2^/g	k^900^, s^−1^	k_s_^900^, s^−1^m^−2^
	Glucose,				
G-7	7	α-Mo_2_C, η-MoC, C	56	1.25	0.05
	Hydroquinone				
H-4	4	α-Mo_2_C, γ-MoC	25	2.30	0.10

* S_a_, specific surface area after conversion; k_900_, constant rate calculated at 900 °C; k_s_^900^, constant rate related to surface area of catalyst.

**Table 3 nanomaterials-11-00873-t003:** Phase composition and BET surface area of molybdenum carbide with different liquid carbon sources.

Carbon Source	Glucose	Hydroquinone	Ascorbic Acid [32]	Sucrose [27]	Urea [26]
[R]/[Mo]	4–7	4	0.6–5	2–3.1	1–7
Phase composition	α-Mo_2_C; η-MoC; C	α-Mo_2_C; γ-MoC; C	β-Mo_2_C; η-MoC; C	Mo_2_C MoC_1−_x	γ-Mo_2_N; α-Mo_2_C
S_a_ (BET), m^2^/g	7.1–175.0	203.0	1.4–63.0	5	12.0–22.0

## Data Availability

The data presented in this study are available on request from the corresponding author.

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
