# Peer review of "Simple Synthesis of Molybdenum Carbides from Molybdenum Blue Nanoparticles"

_nanomaterials, 2021, doi:10.3390/nano11040873_

Round 1

Reviewer 1 Report

title of the manuscript should be more relevant, representative.

Please be sure that your manuscript thoroughly establishes how this work is original. Specific comparisons should be made to previously published materials that have a similar purpose. Please indicate (in the manuscript) a strong case for how this study is a major advance.

The language of the manuscript must be significantly improved so that it is easy to read. You need to correct the grammar. Please go through the entire manuscript and shorten and correct some sentences.

Abstract. This section should be revised by inserting some key results, while keeping only the relevant information/data (i.e. delete the first sentence, introduce the characterization techniques, significant results, etc.). Include your recommendations and future prospects.

Introduction. The significance of this study should be more emphasize in the introduction. The aim must be specially highlighted to attract more readers.

Materials. The details regarding the used chemicals are provided (concentration, purity, etc.) should be presented. Synthesis. Expression like “certain amount” are confusing. There is no quantitative chemical analysis to support the [R]/[Mo] ratio.

Results and discussions. The quality and interpretation of obtained results should be improved. Formatting and quality of the figures should be considerably improved (the plots are not of the same size, some of them are too large. Figure 3. Please define A on the x axis. Please replace “,” with “.” when refer to numbers. Figure 8. Please overlap the bar scale on the figure and remove the bottom part.

Despite the interest for the molybdenum carbides presented in the manuscript, rigorous analysis of characterizations is required, the obtained results should be supported/ compared by previous results and conclusions drawn from results have to be clearly separated from suggested interpretation.

I consider that the article can be accepted for publication only after a major, major revision.

Author Response

Dear Reviewer, thank you for your careful reading of the manuscript. Here are the answers to your comments:

  1. Title of the manuscript should be more relevant, representative.                  -Another manuscript title suggested (line 2-3).
  2. Please be sure that your manuscript thoroughly establishes how this work is original. Specific comparisons should be made to previously published materials that have a similar purpose. Please indicate (in the manuscript) a strong case for how this study is a major advance.                                        -Added information about the relevance of this study, about the significance of the results (Section 1. Introduction, line 171-175, 181-191, 200-204) Added comparison with literature data (Section 4, line 1397-1402, 1416-1419).
  3. The language of the manuscript must be significantly improved so that it is easy to read. You need to correct the grammar. Please go through the entire manuscript and shorten and correct some sentences.                         -English editing done (MDPI English Editing Services was used).
  4. Abstract. This section should be revised by inserting some key results, while keeping only the relevant information/data (i.e. delete the first sentence, introduce the characterization techniques, significant results, etc.). Include your recommendations and future prospects.                          -Abstract was corrected according to reviewers recommendations (line 12-24).
  5. Introduction. The significance of this study should be more emphasize in the introduction. The aim must be specially highlighted to attract more readers.                                                                                                           -The introduction has been revised, information has been added showing the relevance and significance of the research being carried out (Section 1. Introduction, line 171-175, 181-191, 200-204), the purpose of this work is emphasized (Section 1, line 200-204).
  6. Materials. The details regarding the used chemicals are provided (concentration, purity, etc.) should be presented. Synthesis. Expression like “certain amount” are confusing. There is no quantitative chemical analysis to support the [R]/[Mo] ratio.                                                                         -The Materials has been revised, details of synthesis procedure were added to manuscript (Section 2.1, line 409-410 and 2.2, line 417 - 424).
  7. Results and discussions. The quality and interpretation of obtained results should be improved. Formatting and quality of the figures should be considerably improved (the plots are not of the same size, some of them are too large. Figure 3. Please define A on the x axis. Please replace “,” with “.” when refer to numbers. Figure 8. Please overlap the bar scale on the figure and remove the bottom part.                                                               -The quality and interpretation of obtained results were improved.
  8. Despite the interest for the molybdenum carbides presented in the manuscript, rigorous analysis of characterizations is required, the obtained results should be supported/ compared by previous results and conclusions drawn from results have to be clearly separated from suggested interpretation.                                                                               -The obtained results were compared by previous results and conclusions (Section 4, line 1397-1402, 1416-1419).

Reviewer 2 Report

comments attached.

Author Response

Dear reviewer, thank you for your careful reading of the manuscript. Here are the answers to your comments:

There is some new information in this paper and a form can probably be published. My biggest complaint is that there are various samples and surface areas for samples reported but I could not figure out how different treatments led to different surface areas. There was no information on how the surface areas and structures evolve with temperature treatments. It was very hard to get much more information out of this other than the fact that one can treat molybdena salts with hydroquinone and form carbides.

In this work, the treatment of molybdenum salts with glucose or hydroquinone was not carried out. The idea of ​​the synthesis was that the interaction of ammonium heptamolybdate with an organic reducing agent in an acidic medium resulted in the formation of a dispersion of molybdenum blue nanoparticles. It was these particles that were later used to obtain molybdenum carbides.

The dispersion medium of dispersions contains an excess of a reducing agent and its oxidation products. These compounds serve as a source of carbon during further heat treatment, because it was carried out in an inert environment.

Thus, when molybdenum blue xerogels are heat treated, molybdenum dioxide and carbon are formed, which then form molybdenum carbide. The porous structure of the resulting material is formed in this process.

 1. The information in Table 1 is important but I could not figure out what samples these refer to or how to relate these data to other data presented in the paper. It would be really useful to have a table that just referred to the samples so that one could match up SEM images to surface areas. Furthermore, since the surface areas are very important, it would be useful to know how stable these are. Do they change with heating to various temperatures? What about after reaction?

1. For the convenience of perceiving information, the designations of the samples were introduced. Surface areas are  stable  and didn`t change with heating to 1000 ºC.

After the reaction, a decrease in the specific surface area of the sample is observed due to the removal of free carbon, which is present in the composition of the samples.

2. It is a minor point but I would not refer to the carburization of MoOx as “Temperature Programmed Reduction”. The term TPR is typically used to describe method for characterizing the reducibility of a catalyst.

2. The carburization method is designated as temperature programmed reaction TPRe.

3. The XPS of precursors really do nothing more than lengthen the paper unnecessarily. What information is gained here? The XPS of any molybdena precipitate would look similar. 

3. XPS Spectra removed from the manuscript

4. Line 168: “defenition". I am not actually sure what the authors are trying to say in this sentence; so, I do not know what this word should be.

4. Word corrected – determination

5. A paragraph should always have more than one sentence. (line 232)

5.  Corrected

6. Figure 8 should be placed in a more “journal-quality” form.

6. Corrected

7. I do not believe it is necessary to show adsorption isotherms like that in Figure 9. A table with surface areas and pore sizes is sufficient. Adsorption is not spelled “Adsorbtion”.

7. Adsorption isotherms provide information on the type of pores present (micro-, meso-, macro) and the shape of mesopores (in the form of a hysteresis loop). The authors would like to leave the isotherms and their description in the manuscript.

8. The conversions (these are not rates) shown in Figure 11 are not really meaningful unless one has something to benchmark them to. At very least, the authors should report space times, etc. Again, having a “name” for the samples would be useful for the reader to try to figure out what samples the data was taken on. Table 2 is pointless since these “rates” were not measured at differential conditions.

8. The activity of the samples in the work was compared under the same conditions - at the same contact time (its value was added to the conversion data).

The authors agree with the comment that the values of the rate constant can only be determined under differential conditions. Our experiments were carried out under integral conditions, in which case we can speak of the initial reaction rate. Сhanges were made to the text of the manuscript.

Round 2

Reviewer 1 Report

Section 2.2. Please merge 108-113 lines in a single paragraph.

Line 268 please correct “mkm”

Figures 5 and 6 – the title of x-axis should be “Intensity, a.u.”, without abbreviation.

The following comment was discarded: „Despite the interest for the molybdenum carbides presented in the manuscript, rigorous analysis of characterizations is required, the obtained results should be supported/ compared by previous results and conclusions drawn from results have to be clearly separated from suggested interpretation”. Please introduce the Conclusions section in the manuscript.

It is very important to provide a consistent/ similar data set for all the samples investigated. Example: Why Figure 7-9 and the corresponding text present only for G5 and H4? Why Figure 7 display different magnification?

Results and Discussion: The comparison with previous studies is difficult to follow. There is plenty of general information in these sections. Lines 337and 343 are confusing.

Finally, I consider that the paper is not proper for publication in the present format and still requires revision. Nevertheless, the efforts of performing all the experiments have been significant and I hope that in the near future all the issues will be solved.

Author Response

The authors thank the reviewer for the comments.

1. Section 2.2. Please merge 108-113 lines in a single paragraph.

1. Corrections were done

2. Line 268 please correct “mkm”

2. Corrected.

3. Figures 5 and 6 – the title of x-axis should be “Intensity, a.u.”, without abbreviation.

3. The figures were improved.

4. The following comment was discarded: „Despite the interest for the molybdenum carbides presented in the manuscript, rigorous analysis of characterizations is required, the obtained results should be supported/ compared by previous results and conclusions drawn from results have to be clearly separated from suggested interpretation”. Please introduce the Conclusions section in the manuscript.

4. The obtained results were compared by our previous results and by results from another studies. The sections “Results” and “Discussion” were improved (line 211 – 21, 231 – 232, 314 – 316, 359 – 360, 368 - 378).

The section “Conclusions” was introduced. (line 391 – 406)

5. It is very important to provide a consistent/ similar data set for all the samples investigated. Example: Why Figure 7-9 and the corresponding text present only for G5 and H4? Why Figure 7 display different magnification?

5. SEM-images, nitrogen adsorption isotherms and pore size distributions were presented for G-5 and H-4 because these plots are very similar for all investigated samples of molybdenum carbides (line 271 – 273; 279 – 281; 302 – 303). Also, Reviewer 2 suggested not to put nitrogen adsorption isotherms in our manuscript at all.

Figure 7 was improved.

6. Results and Discussion: The comparison with previous studies is difficult to follow. There is plenty of general information in these sections. Lines 337and 343 are confusing.

6. The comparison with previous studies in section “Results” was introduced (line 211 – 21, 231 – 232, 314 – 316).

The section “Discussion” was improved and now includes Table 3 and comparative analysis of the properties of molybdenum carbides, synthesized using different liquid phase carbon sources (line 359 – 360, 368 - 378).

Reviewer 2 Report

publish as is.

Author Response

The authors thank the reviewer for the comments.

Round 3

Reviewer 1 Report

The manuscript has been substantially improved comparing to the previous ones. The authors have adequately replied to my comments. For me it's OK.